# FourierAugment: Frequency-Based Image Encoding for Resource-Constrained Vision Tasks

## Abstract

Resource-constrained vision tasks, such as image classification on low-end devices, put forward significant challenges due to limited computational resources and restricted access to a vast number of training samples. Previous studies have utilized data augmentation that optimizes various image transformations to learn effective lightweight models with few data samples. However, these studies require a calibration step for optimizing data augmentation to specific scenarios or hardly exploit frequency components readily available from Fourier analysis. To address the limitations, we propose a frequency-based image encoding method, namely FourierAugment, which allows lightweight models to learn richer features with a restrained amount of data. Further, we reveal the correlations between the amount of data and frequency components lightweight models learn in the process of designing FourierAugment. Extensive experiments on multiple resource-constrained vision tasks under diverse conditions corroborate the effectiveness of the proposed FourierAugment method compared to baselines.

## 1 Introduction

As the application areas of computer vision expand, research on lightweight models under constrained conditions such as image classification in mobile or embedded environments with small-scale training datasets is drawing increasing attention (Tao et al. (2020); Lin et al. (2020); Mehta & Rastegari (2022); Girish et al. (2023)). Contemporary studies have designed data augmentation methods to handle the data deficiency problem (Cubuk et al. (2019; 2020); Cheung & Yeung (2022)) or lightweight models to manage limited computational resources (Howard et al. (2017); Tan & Le (2019)). However, conventional data augmentation methods require an optimization process for optimal performance and lightweight models still demand large-scale training datasets for enhanced performance.

In this paper, we focus on the scenario where both the training data and computational resources are constrained. Further, we propose data augmentation as a straightforward method to substantially improve the performance of lightweight models with a restricted amount of training samples. Behind the motivation of our approach rests the special property of the resource-constrained vision tasks. The tasks deal with exceptionally small datasets that reflect the practicality of real-world scenarios (Zhang et al. (2021); Zhou et al. (2022); Peng et al. (2022)). Due to this property, lightweight models with limited training data samples tend to preferentially learn low-frequency related features (Rahaman et al. (2019))—restricting the models from equipping with discriminative features. To cope with this limitation, we design a frequency-based image encoding method, FourierAugment.

The proposed FourierAugment method explicitly provides the information from various frequency bands to a lightweight model by exploiting the discrete Fourier transformation (DFT). Consequently, the model with FourierAugment learns both low- and high-frequency information in balance (Fig. 1)—resulting in richer features promoting boosted performance. Besides, the proposed FourierAugment method is easily applicable to existing models without complicating the model architecture or increasing the amount of computation.

In the process of designing FourierAugment, we have devised a thorough empirical study—shedding light on the model design process. We reveal the correlation between the amount of data and fre-

(a) Original  (b) FourierAugment

Figure 1: Comparison of the learning process with FourierAugment against that with the original data, in resource-constrained conditions. (a) the process of learning with the original data; the model tends to learn only one type of features. (b) the process of learning with the proposed FourierAugment method; the model learns features from various frequency bands.

quency components lightweight models learn; it is challenging for lightweight models to learn high-frequency related features in the early stage of learning because humans prefer low-frequency information and provide ground-truth labels for learning (Wang & Raj (2017)). Furthermore, we have performed comprehensive comparative experiments to establish the effectiveness of the proposed FourierAugment method both qualitatively and quantitatively. As a result of our study, we have achieved new state-of-the-art performance.

In summary, our main contributions are as follows:

- **FourierAugment**: We propose FourierAugment to significantly improve the performance of lightweight models without complicating the model architectures.
- **Empirical Study**: We reveal the correlation between the number of data samples and learned frequency components of lightweight models.
- **SoTA Performance**: As a result of our study, we achieve new state-of-the-art performance for several public benchmarks.
- **Open Source**: To contribute to the research society, we make the source code of the proposed FourierAugment method and related dependencies public.

## 2 RELATED WORK

### 2.1 IMAGE TRANSFORMATION AND DATA AUGMENTATION

**Image transformation** varies the original image through multiple operations. Image transformation largely encompasses two classes of approaches (Laganiere (2014); Szeliski (2022)): 1) changing pixel values and 2) moving the position of pixels. First, contrast, color, sharpness, brightness, etc. exemplify image transformation methods changing pixel values. Next, image transformation moving the position of pixels includes rotation, shearing, translation, zooming, and warping.

**Data augmentation** for vision tasks, in general, combines image transformation to handle the problem of limited data—enhancing generalization (Shorten & Khoshgoftaar (2019)). Zhang et al. (2017) proposed MixUp that augments new data by mixing two data and labels in a specified ratio. CutMix (Yun et al. (2019)) takes a cut-and-paste method and fills a part of an image with a patch of another image. AugMix (Hendrycks et al. (2019)) remixes the original image with the augmented data to prevent the augmented image from changing its manifold. AutoAugment (Cubuk et al. (2019)) automatically searches for improved data augmentation policies. Fast AutoAugment (Lim et al. (2019)) complements the slow optimization process of AutoAugment and RandAugment (Cubuk et al. (2020)) requires only two hyper-parameters to perform data augmentation. Deep AutoAugment (Zheng et al. (2022)) reduces strong human priors and performs data augmentation search in a fully automated manner. Nonetheless, this class of data augmentation demands a complicated optimization process and specific domain knowledge.

**Frequency-based data augmentation**—employing readily accessible features from Fourier analysis—has mainly drawn attention from the medical image processing society. For example, a research team has proposed a domain generalization method that only augments the amplitude information during training (Xu et al. (2021)); domain shift barely affects the phase information for

medical images. Next, a study for differentiating diabetes from non-diabetes with foot temperature photographs has utilized the Fourier transform as a way to make up for insufficient data (Anaya-Isaza & Zequera-Diaz (2022)). Another study (Yang et al. (2022)) has designed a source-free domain adaptation method for medical image segmentation. Nevertheless, these Fourier-based methods tend to fail in generalizing over multiple tasks since they are specifically designed for certain task domains.

## 2.2 Frequency Domain in Computer Vision

Recent studies have demonstrated that frequency domain features are effective in classification and few-shot learning tasks (Xu et al. (2020); Chen & Wang (2021)). These studies have applied the discrete cosine transformation (DCT) to convert images from the spatial domain to the frequency domain. This simple feature conversion has resulted in classification performance improvement by approximately 1∼2% (Xu et al. (2020)). Furthermore, processing frequency domain features and spatial domain features separately and integrating the two types of features in the later stage has boosted the performance of few-shot learning (Chen & Wang (2021)).

Further, researchers have investigated the relationship between features from different frequency bands and CNN models. First, it is a well-known fact that low-frequency components (LFC) provide key information to CNNs. LFC corresponds to broader patterns and textures in space. LFC represents the approximate shape, boundaries, and major features of the subject (Bharati et al. (2004)). Indeed, CNNs show a tendency to learn LFC first than high-frequency components (HFC) (Rahaman et al. (2019)), which is explained by the fact that humans supply the labels for target datasets (LFC has more critical effects on the human visual system than HFC) (Wang et al. (2020)). On the other hand, HFC, which seemingly contains lots of noise, provides detailed information to CNN models. LFC helps CNN models achieve robustness on random perturbations (Xu (2018); Xu et al. (2019); Yin et al. (2019)) and HFC is associated with finer details and small features (Yin et al. (2019); Wang et al. (2020)). In fact, HFC learned at the later stage of learning aids CNN models in generalizing over training and validation datasets (Wang et al. (2020); Zhao et al. (2022)).

## 2.3 Resource Constrained Vision Tasks

As the scope of application of AI increases, lightweight models have appeared to solve computer vision tasks in low-end environments such as mobile devices and IoT. We present the resource-constrained environments reported in the literature and introduce resource-constrained vision tasks.

**Resource-Constrained Environments**. Resource-constrained environments fall into several categories depending on the memory usage and computational time (Bianco et al. (2018)). Specifically, the resource-constrained conditions are as follows:

- **Memory usage**: high ($\leq$ 1.4GB), medium ($\leq$ 1.0GB) and low ($\leq$ 0.7GB)
- **Computational time**: half real-time ($\leq$ 66ms), real-time ($\leq$ 33ms), and super real-time ($\leq$ 17ms)

The NVIDIA Jetson series, a representative product of embedded systems, has 64GB or less of internal storage. To use only the built-in storage, we should limit the size of the dataset.

**Image Classification**. A large number of researchers have studied lightweight models for image classification, a representative computer vision task. SqueezeNet (Iandola et al. (2016)) uses a combination of $1 \times 1$ convolutional filters, which reduce the number of input channels, and fire modules, which act as a lightweight alternative to traditional convolutional layers. MobileNet (Howard et al. (2017)) involves the depthwise separable convolution structure to optimize the size of the model. MCUNet (Lin et al. (2020)) is a lighter model than the two models and can run on a microcontroller; MCUNet extracts weights from the layers during training and increases sparsity by removing parts of the weights. Moreover, scalable model architectures such as ResNet (He et al. (2016)) and EfficientNet (Tan & Le (2019)) have emerged.

**Few-Shot Class-Incremental Learning**. Recently proposed few-shot class-incremental learning (FSCIL) (Tao et al. (2020)) studies the Class-incremental learning task with extremely few training samples (less than 10 samples per class) in incremental sessions; a learning method receives abundant object classes and data in the base session. The earliest work on FSCIL has proposed a neural

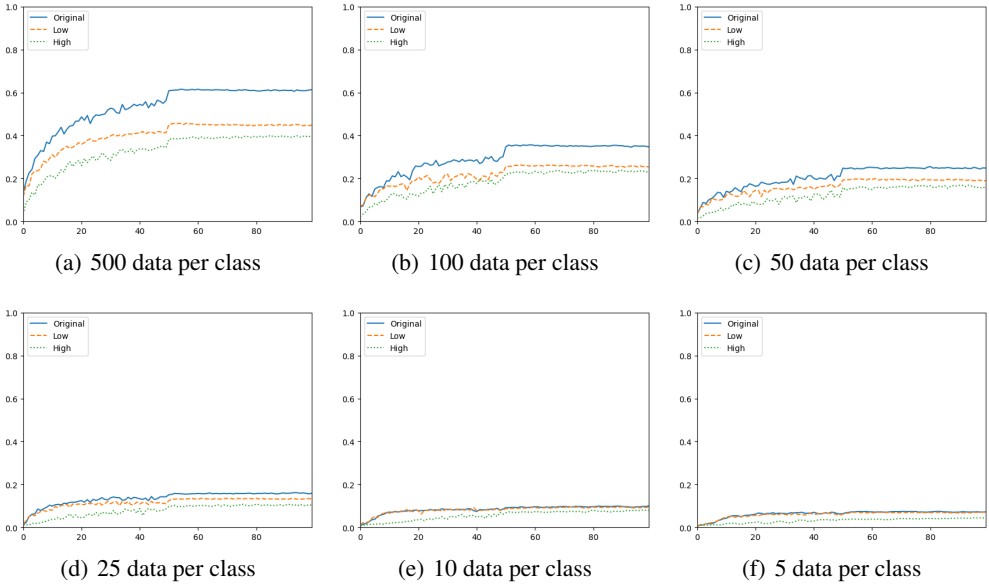

Figure 2: The effect of the number of data per class and the input frequency components on accuracy. The horizontal and vertical axes represent the epochs and the accuracy, respectively. The blue solid line is the accuracy of the model learned from the original images. The orange dotted line and the green dotted line are the accuracies of the models learned with LFC images and HFC images, respectively. As the number of training data decreases, the accuracy of the original image model and that of the LFC model becomes indistinguishable.

gas network to maintain the topology of features across previous and new object classes (Tao et al. (2020)). CEC has attempted to learn context information for classifiers utilizing a pseudo incremental learning paradigm and a graph neural network (Zhang et al. (2021)). FACT has focused on forward compatibility and secured room for incremental classes in the embedding space by utilizing virtual prototypes and virtual instances (Zhou et al. (2022)). ALICE has encouraged the model to learn discriminative features through the angular penalty loss (CosFace) (Wang et al. (2018)) as well as transferable representations by class augmentation and data augmentation techniques (Peng et al. (2022)).

## 3 EMPIRICAL OBSERVATION

Our goal is to improve the performance of neural models by exploiting the frequency analysis in two limited conditions: the amount of data and the size of the models. Previous studies (Xu (2018); Rahaman et al. (2019); Xu et al. (2019); Yin et al. (2019); Wang et al. (2020)) have presented frequency characteristics of CNN models, but there has been no research from a frequency perspective on features of constrained conditions. Therefore, it is necessary to study what frequency component models learn in these conditions. In this section, we describe our research hypothesis and empirical validation.

### 3.1 RESEARCH HYPOTHESIS

We investigate how the amount of data and the size of models affect the frequency components models learn. Previous studies have revealed CNN models prefer to learn LFC first because they use human-classified datasets (Rahaman et al. (2019); Wang et al. (2020)). This leads to speculation that models would preferentially learn LFC —resulting in insufficiently learned HFC— when we constrain the amount of data and the size of models. We state our hypotheses as follows:

- **Hypothesis**: When the sizes of datasets and models become restricted, models would mainly learn LFC rather than learn LFC and HFC in a balanced manner.

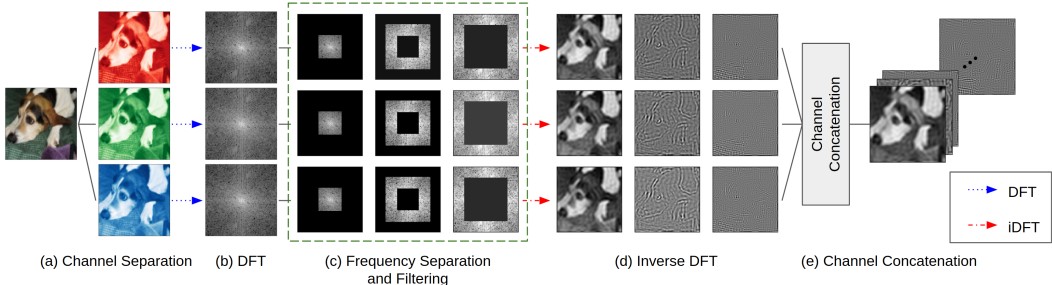

(a) Channel Separation     (b) DFT     (c) Frequency Separation and Filtering     (d) Inverse DFT     (e) Channel Concatenation

Figure 3: The overall data processing pipeline of FourierAugment. FourierAugment separates the RGB channels, applies discrete Fourier transformation (DFT) to each channel, processes low-, mid-, and high-frequency components, and concatenates the processed components as independent channels.

## 3.2 EMPIRICAL VALIDATION[1]

**Settings**. We use ResNet18 (He et al. (2016)) and *mini*ImageNet (Russakovsky et al. (2015)) as the baseline. To verify which frequency component the model learns, we use three types of images: original images, LFC images (containing only LFC), and HFC images (consisting of only HFC). We limit the training data per class to 500, 100, 50, 25, 10, and 5 to examine what frequency components the model learns when we constrain the amount of data, on the small model. The total number of classes is 100.

**Results and Analysis**. Fig. 2 shows that as the number of data per class diminishes, the difference of accuracy between the model learned with the LFC images and that learned with the original images decreases. In particular, when the number of data is reduced to less than 10, the accuracy of the two models becomes nearly identical—analogous training curves indicating the two models learn the same category of frequency components (Wang et al. (2020)). On the other hand, the HFC image model differs in performance from the original image model by $1.95\%$ when the number of data is 10 and by $2.66\%$ when the number of data is 5. In this experiment, we can observe that lightweight models prefer to learn LFC rather than HFC given a small number of data.

## 4 METHODOLOGY

### 4.1 MOTIVATION

We propose FourierAugment to address the problem of poorly learning HFC in resource-constrained conditions. Previous studies have analyzed the influence of each frequency component by separating frequencies using the Fourier transform (Rahaman et al. (2019); Wang et al. (2020))—in particular, learning HFC properly is crucial for performance. However, previous studies have only utilized one frequency domain at a time, either LFC or HFC (Rahaman et al. (2019); Wang et al. (2020)). Furthermore, they have not presented methods to learn various frequency components. We propose a simple but effective method of integrating images separated by low- and high-frequency bands into a single data to shift the attention of resource-constrained models toward HFC.

### 4.2 FOURIERAUGMENT

The proposed FourierAugment is a frequency-based image encoding that helps models learn each frequency component better by explicitly feeding a set of distinct frequency components separated by bands. Fig. 3 depicts the data processing pipeline of the proposed FourierAugment method. First, we separate the RGB channels of the input image. Then, we perform the discrete Fourier transformation (DFT) to convert each channel image into the frequency domain from the spatial domain. We maintain the magnitude spectrums of the converted channels in the frequency domain. Next, we define a filter bank as follows:

---

[1]For more detailed description and results, refer to the supplementary materials.

Table 1: Top-1 accuracies of RA, DAA, and FA on the *mini*ImageNet and ImageNet dataset

| Method | *mini*ImageNet | | ImageNet | | |
|---|---|---|---|---|---|
| | 100 | 500 | 100 | 500 | full |
| baseline | 35.60 | 61.52 | 19.13 | 57.81 | 65.16 |
| +AM | 34.42 | 58.89 | 36.22 | 59.15 | 67.38 |
| +RA | 35.98 | 61.94 | 22.61 | 51.99 | 67.33 |
| +DAA | 35.81 | 64.30 | 22.23 | 57.68 | 67.50 |
| +FA | **37.35** | **65.23** | **37.26** | **61.44** | **67.82** |

$$f_i = \{(x,y)| \frac{I}{2n}(i-1) \le |x| \le \frac{I}{2n}i, \frac{I}{2n}(i-1) \le |y| \le \frac{I}{2n}i\}, \tag{1}$$

where $i (= 1, ..., n)$ and $n$ represent the index of each filter and the number of total filters in the filter bank, respectively. We apply these filters to the magnitude spectrums, invert the filtered magnitude spectrums into the spatial domain, and concatenate the inverted signals. As a result, the total number of channels becomes $n \times 3$. We generally set $n$ as 2 or 3—moderately separating frequency bands can benefit from separation without compromising the performance; The dense separation results in lower performance despite the increased input size.

After FourierAugment, the resulting input image becomes $n \times 3$ channels. The change in the input shape requests modification of the backbone architecture; the input layer should receive a ($n \times 3$)-channeled input rather than 3-channeled input. Moreover, FourierAugment functions as a few convolutional layers since it extracts a set of features. This functionality creates synergy when the first few layers of the backbone get omitted (Xu et al. (2020)). In the case of ResNet (He et al. (2016)), skipping the first $7 \times 7$ convolutional layer provokes performance optimization.

## 5 EXPERIMENTS[2]

The proposed FourierAugment method can improve the performance of lightweight models with fewer data. First, we demonstrate the effectiveness of FourierAugment in resource-constrained vision tasks: image classification, and FSCIL. For comparative study, we employ high-performance data augmentation methods as baselines because it is rare for other image transformations to be used alone. Specifically, we use AugMix (AM) (Hendrycks et al. (2019)), RandAgument (RA) (Cubuk et al. (2020)) and Deep AutoAugment (DAA) (Zheng et al. (2022)), the state-of-the-art methods of ImageNet; Existing methods using frequency are designed to suit specific data and are not suitable for general tasks. Further, we verify that FourierAugment indeed helps models learn HFC through quantitative and qualitative studies.

### 5.1 RESOURCE CONSTRAINED VISION TASKS

#### 5.1.1 IMAGE CLASSIFICATION

**Settings**. We utilize two sizes of datasets to examine the effect of FourierAugment over the size of datasets. Concretely, we employ ImageNet 1K (Russakovsky et al. (2015)) and *mini*ImageNet as large and small datasets, respectively. ImageNet 1K includes 1,000 classes and approximately 1,000 images per class and *mini*ImageNet, a subset of ImageNet, encompasses 100 classes and 500 images per class. To evaluate the effect of the amount of data per class, we conduct additional experiments using subsets of ImageNet with 500 and 100 images per class, and subsets of *mini*ImageNet with 100 images per class. This examines the impact of both the total amount of data and the amount of data per class. ImageNet offers images with the resolution of $224 \times 224$, and *mini*imageNet with $84 \times 84$. Moreover, we use ResNet18 as a base model.

---

[2]For more detailed description and results, refer to the supplementary materials.

Table 2: Comparative study results of the proposed FourierAugment method on the *mini*ImageNet dataset.

| Method | Top-1 Accuracy in each session (%) | | | | | | | | | PG |
|--------|-------|-------|-------|-------|-------|-------|-------|-------|-------|--------|
|        | 0 | 1 | 2 | 3 | 4 | 5 | 6 | 7 | 8 |  |
| CEC | 78.00 | 72.89 | 69.01 | 65.45 | 62.36 | 59.09 | 56.42 | 54.28 | 52.63 | - |
| + AM | 73.45 | 68.46 | 64.06 | 60.59 | 57.41 | 54.32 | 51.40 | 49.32 | 47.42 | -5.21 |
| + RA | 76.80 | 71.62 | 67.17 | 63.83 | 60.54 | 57.51 | 54.90 | 52.93 | 51.36 | -1.27 |
| + DAA | 76.80 | 71.62 | 67.26 | 63.63 | 60.59 | 57.71 | 54.96 | 52.81 | 51.57 | -1.06 |
| + FA | 80.30 | 74.34 | 69.94 | 66.48 | 63.37 | 60.63 | 57.59 | 55.45 | **53.77** | **+1.14** |
| FACT | 75.92 | 70.62 | 66.29 | 62.79 | 59.46 | 56.27 | 53.23 | 51.05 | 49.20 | - |
| + AM | 74.73 | 69.68 | 65.14 | 62.01 | 59.08 | 56.29 | 53.52 | 51.72 | 50.00 | + 0.80 |
| + RA | 77.47 | 72.22 | 68.10 | 64.45 | 61.37 | 58.64 | 55.49 | 53.58 | 52.02 | +2.97 |
| + DAA | 78.73 | 73.20 | 68.77 | 65.05 | 62.16 | 59.11 | 55.89 | 53.94 | 52.49 | +3.44 |
| + FA | 81.25 | 75.86 | 71.50 | 67.68 | 64.50 | 61.05 | 57.84 | 55.82 | **54.01** | **+4.96** |
| ALICE | 81.03 | 72.48 | 68.94 | 65.15 | 62.68 | 60.11 | 57.74 | 56.85 | 55.72 | - |
| + AM | 77.53 | 69.03 | 64.39 | 61.21 | 57.95 | 55.18 | 52.88 | 51.29 | 49.62 | -6.10 |
| + RA | 77.38 | 69.02 | 65.40 | 61.84 | 59.52 | 56.78 | 54.50 | 52.55 | 51.47 | -4.25 |
| + DAA | 76.53 | 67.97 | 64.24 | 60.21 | 57.77 | 55.41 | 53.00 | 51.57 | 50.94 | -4.78 |
| + FA | 80.88 | 73.06 | 69.57 | 65.80 | 63.46 | 60.61 | 58.32 | 57.15 | **56.09** | **+0.37** |

**Results and Analysis**. Table 1 shows the result of applying each method to *mini*ImageNet and ImageNet. In *mini*ImageNet experiments, when the number of images per class was 500, Fourier-Augment improved performance by 3.71% over the baseline. When the number of images per class was 100, FourierAugment improved performance by 1.75% over the baseline. On *mini*ImageNet, FourierAugment outperforms other data augmentation methods as well.

In ImageNet, FourierAugment exhibits superior performance than the baseline data augmentation methods both when restricting the number of data and when not. Accuracies improved by 18.13%, 3.63%, and 2.66%, respectively, when the numbers of data per class were 100, 500, and whole. Especially, in the ImageNet-100 experiments, the most constrained experiments, other methods improved performance by 3.10% and 3.45%, respectively, while FourierAugment improved performance by 18.13%, which is a significant margin. Therefore, the smaller the number of data, the higher the effect of FourierAugment. We note that the full data case is not the resource-constrained environment—the focus of our work—and this result agrees with our expectation since we specifically designed FourierAugment for resource-constrained vision tasks, not for data-abundant cases; although FourierAugment enhances the performance for the full data case as well.

### 5.1.2 FSCIL

**Settings**. We employ *mini*ImageNet and CUB200 (Wah et al. (2011)) datasets for FSCIL experiments. *mini*ImageNet includes 100 object classes with 600 images for each class, totaling 60,000 images of the $84 \times 84$ size. We split the 100 classes into 60 base session classes and 40 incremental session classes following the convention (Tao et al. (2020)). CUB200 originally designed for fine-grained image classification offers $224 \times 224$ sized 11,788 images of 200 object classes. We divide the 200 object classes into 100 base session classes and 100 incremental session classes as the standard evaluation protocol (Tao et al. (2020)). *mini*ImageNet follows the 5-way 5-shot setting and CUB200 does the 10-way 5-shot setting.

We investigate the effectiveness of the proposed FourierAugment method and other data augmentations on the following FSCIL models: continually evolved classifiers (CEC) (Zhang et al. (2021)), forward compatible training (FACT) (Zhou et al. (2022)), and augmented angular loss incremental classification (ALICE) (Peng et al. (2022)). We do not make much change to the existing models to focus only on the influence of our proposed method. Specifically, we kept the training procedure the same but removed the first convolutional layer as mentioned before.

**Metrics**. We report the Top-1 accuracy after each session denoted as $\mathcal{A}_i$, where $i$ stands for the session number. Furthermore, we quantitatively measure the performance gain ($PG$) owing to the

Table 3: Comparative study results of the proposed FourierAugment method on the CUB200 dataset.

| Method | Top-1 Accuracy in each session (%) | | | | | | | | | | | PG |
|---|---|---|---|---|---|---|---|---|---|---|---|---|
| | 0 | 1 | 2 | 3 | 4 | 5 | 6 | 7 | 8 | 9 | 10 | |
| CEC | 75.85 | 71.94 | 68.50 | 63.50 | 62.43 | 58.27 | 57.73 | 55.81 | 54.83 | 53.52 | 52.28 | - |
| + AM | 74.73 | 70.30 | 66.11 | 60.83 | 60.30 | 55.90 | 54.19 | 52.41 | 51.74 | 49.91 | 48.15 | -4.13 |
| + RA | 73.22 | 69.29 | 64.71 | 60.04 | 59.31 | 55.44 | 54.32 | 51.90 | 51.21 | 49.59 | 48.10 | -4.18 |
| + DAA | 74.15 | 70.06 | 66.08 | 60.88 | 60.00 | 56.59 | 54.95 | 52.69 | 51.52 | 50.24 | 49.09 | -3.19 |
| + FA | 79.94 | 75.22 | 70.89 | 66.05 | 64.79 | 61.37 | 60.27 | 58.20 | 57.22 | 56.31 | **55.09** | **+2.81** |
| FACT | 75.90 | 73.23 | 70.84 | 66.13 | 65.56 | 62.15 | 61.74 | 59.83 | 58.41 | 57.89 | 56.94 | - |
| + AM | 79.21 | 74.73 | 71.25 | 66.85 | 66.09 | 63.40 | 62.65 | 61.50 | 59.63 | 58.82 | 57.81 | +0.87 |
| + RA | 79.30 | 75.18 | 71.28 | 66.35 | 66.06 | 62.20 | 60.55 | 59.37 | 58.33 | 57.35 | 56.13 | -0.81 |
| + DAA | 79.97 | 75.90 | 72.43 | 67.09 | 66.55 | 62.93 | 61.08 | 59.87 | 59.00 | 57.56 | 56.14 | -0.80 |
| + FA | 79.80 | 74.30 | 70.88 | 66.58 | 66.27 | 62.77 | 62.61 | 61.44 | 59.13 | 58.92 | **57.87** | **+0.93** |
| ALICE | 78.14 | 73.15 | 70.64 | 67.33 | 65.57 | 62.88 | 62.05 | 61.09 | 59.82 | 59.79 | 59.27 | - |
| + AM | 78.46 | 73.91 | 71.22 | 67.97 | 65.91 | 63.06 | 62.25 | 61.41 | 59.70 | 59.56 | 58.92 | -0.35 |
| + RA | 77.48 | 72.89 | 70.61 | 67.06 | 65.09 | 62.07 | 61.34 | 60.40 | 58.95 | 58.81 | 58.25 | -1.02 |
| + DAA | 77.30 | 71.94 | 69.24 | 65.93 | 64.35 | 61.56 | 60.84 | 60.01 | 58.61 | 58.48 | 57.84 | -1.43 |
| + FA | 78.53 | 75.12 | 72.71 | 69.34 | 67.55 | 64.86 | 63.89 | 63.18 | 61.72 | 61.76 | **60.84** | **+1.57** |

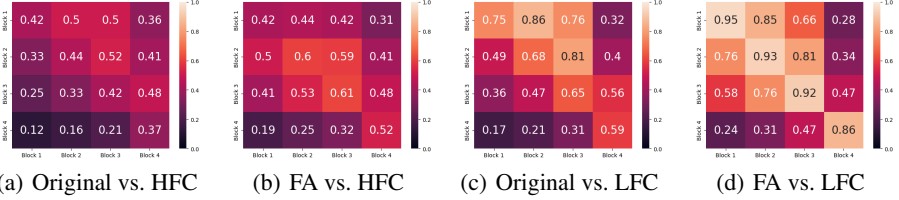

(a) Original vs. HFC     (b) FA vs. HFC     (c) Original vs. LFC     (d) FA vs. LFC

Figure 4: Analysis of the similarity between features from each block of ResNet18 measured by CKA. (a) the similarity between the original model and the HFC model, (b) the similarity between the FourierAugment model and the HFC model, (c) the similarity between the original model and the LFC model, and (d) the similarity between the FourierAugment model and the LFC model.

proposed FourierAugment method, *i.e.*, $PG = \mathcal{A}_N^{\mathrm{DA}} - \mathcal{A}_N^{\mathrm{Orig}}$, where $\mathcal{A}_i^{\mathrm{DA}}$ and $\mathcal{A}_i^{\mathrm{Orig}}$ refer to the Top-1 accuracy with and without data augmentation, respectively and $N$ indicates the last session number.

**Results and Analysis**. Tables 2 and 3 delineate the comparison results on *mini*ImageNet and CUB200, respectively. In all cases, the proposed FourierAugment method enhances the performance of the FSCIL models. AM, RA, and DAA have the effect of improving the performance of FACT on *mini*ImageNet, but less effective than FourierAugment. On other models, AM, RA, and DAA degrade performance. On CUB200, FourierAugment also gained higher performance margins of 2.81%, 0.93%, and 1.57% on three models, respectively.

## 5.2 Effectiveness of FourierAugment

In this section, we verify the effectiveness of FourierAugment through centered kernel alignment (CKA) (Kornblith et al. (2019); Davari et al. (2022)) and visualization of occlusion sensitivity (Zeiler & Fergus (2014)).

### 5.2.1 Centered Kernel Alignment (CKA)

**Settings**. To verify that FourierAugment helps to learn various frequency components, we examine if the CNN model learns both low- and high-frequency features with FourierAugment using CKA; CKA numerically analyzes how similar two sets of features from a pair of models are and represents the similarity as a single value between 0 to 1 (1 signifying the uniformity). For this, we contrast the features at each block of ResNet18 trained with the four data configuration schemes: 1) original data, 2) low- and middle-frequency pass-filtered data (low-frequency components; LFC), 3) high-

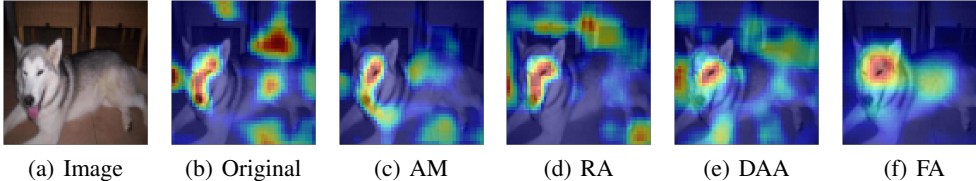

|  |  |  |  |  |  |
|---|---|---|---|---|---|
| (a) Image | (b) Original | (c) AM | (d) RA | (e) DAA | (f) FA |

Figure 5: The areas each model uses for prediction. (a) the source image is used for prediction. (b) the model trained with original images focuses on irrelevant regions to the object. The models trained with (c) AugMix, (d) RandAugment, and (e) Deep AutoAugment still use unrelated areas. (f) the model with FourierAugment concentrates on the object.

and middle-frequency pass-filtered data (high-frequency components; HFC), and 4) low-, middle- and high pass filtered data (FourierAugment).

**Results and Analysis**. Fig. 4 represents the feature similarity analysis results. In Block 1, the similarity between Original and HFC is the same as that between FourierAugment and HFC (Fig. 4(a) and Fig. 4(b)). However, the similarity between FourierAugment and HFC becomes more prominent than that between the Original and HFC in later Blocks—signifying FourierAugment helps the model to learn high-frequency features far more significantly. Besides, the FourierAugment model has a higher similarity to LFC than the original model (Fig. 4(c) and Fig. 4(d)). This result indicates that FourierAugment aids the model in learning not only high-frequency features but also low-frequency features, which is not notable with the original data—attesting to the validity of the effectiveness of FourierAugment.

### 5.2.2 OCCLUSION SENSITIVITY

**Settings**. To verify the effectiveness of FourierAugment, we compare where the models learned with each data augmentation technique focus on for prediction. Occlusion sensitivity presents the areas that models concentrate on. If the heatmap approaches the red spectrum, it indicates the region where the model is primarily attentive, whereas proximity to the blue spectrum suggests the region seldom utilized by the model (Zeiler & Fergus (2014); Van Noord et al. (2015); Aminu et al. (2021)). We train ResNet18 on *mini*ImageNet with four data augmentation methods: AugMix, RandAugment, Deep AutoAugment, and FourierAugment.

**Results and Analysis**. Fig. 5 visualizes the attention of the models learned by different data augmentation methods. The model trained on the original images exhibits a proclivity to attend to regions unrelated to the target object for prediction (Fig. 5(b)). Models employing AugMix, RandAugment, and Deep AutoAugment tend to focus more on the object than the original model, but they still allocate attention to extraneous regions (Fig. 5(c), Fig. 5(d), and Fig. 5(e)). This phenomenon can be attributed to the constraints of dataset size and model size, which preclude the model from achieving complete proficiency in discerning optimal focus areas. In contrast, the model employing Fourier-Augment demonstrates a notable ability to concentrate attention precisely on the object essential for accurate prediction (Fig. 5(f)).

## 6 CONCLUSION

In this paper, we presented FourierAugment, a frequency-based image encoding. By utilizing DFT, FourierAugment provides a balanced representation of low- and high-frequency information, resulting in improved feature richness and discriminability. FourierAugment stands as the inaugural data augmentation that improves the performance of lightweight models by facilitating the learning of various frequency components in balance. Extensive experiments demonstrated the effectiveness of FourierAugment, achieving new state-of-the-art performance on benchmark datasets. Our approach is easily applicable to existing models without architectural complexity or increased computational requirements and is effective on general image data. Overall, FourierAugment offers a practical solution for boosting the performance of lightweight models in resource-constrained environments.

ETHICS STATEMENT

We proposed data augmentation as a straightforward method to substantially improve the performance of lightweight models with an excessively restricted amount of training samples. We believe our research does not raise potential concerns where appropriate, topics include, but are not limited to, studies that involve human subjects, practices to data set releases, potentially harmful insights, methodologies and applications, potential conflicts of interest and sponsorship, discrimination/bias/fairness concerns, privacy and security issues, legal compliance, and research integrity issues (e.g., IRB, documentation, research ethics). We used only public datasets and source codes.

REPRODUCIBILITY STATEMENT

All the missing details for experiments in Section 3 and Section 5 are in the Appendix and the supplementary material. Details of empirical validation and used datasets are in Appendix B. We described the implementation details of classification experiments in Appendix C.1.1, and details of FSCIL experiments are summarized in Appendices C.2.1 and C.2.2. We also attached the source code in the supplementary material.

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

## A    RESOURCE-CONSTRAINED ENVIRONMENTS

Table 4: Memory usage and computational time for each neural network

| Models | Memory usage | | | Computational time | | |
|---|---|---|---|---|---|---|
| | 16 | 32 | 64 | 16 | 32 | 64 |
| AlexNet | 0.84 | 0.92 | 0.99 | 5.39 | 4.77 | - |
| DenseNet-121 | 0.71 | 0.71 | 0.99 | 40.41 | 38.22 | - |
| DenseNet169 | 0.93 | 0.97 | 1.04 | 92.97 | 88.95 | - |
| FBResNet-152 | 0.97 | 1.12 | 1.31 | 94.26 | 97.47 | - |
| GoogLeNet | 1.09 | 1.51 | 2.35 | 19.77 | 19.96 | - |
| MobileNet-v1 | 0.67 | 0.71 | 0.78 | 10.82 | 10.58 | 10.55 |
| MobileNet-v2 | 0.66 | 0.70 | 0.78 | 13.18 | 13.10 | 12.72 |
| ResNet-101 | 1.08 | 1.37 | 1.94 | 58.11 | - | - |
| ResNet-152 | 1.15 | 1.43 | 2.01 | 82.35 | - | - |
| ResNet-18 | 0.71 | 0.75 | 0.89 | 11.99 | 10.73 | 12.45 |
| ResNet-34 | 0.90 | 1.09 | 1.47 | 20.41 | 18.48 | 17.97 |
| ResNet-50 | 0.99 | 1.28 | 1.86 | 35.72 | - | - |
| ShuffleNet | 0.95 | 0.99 | 1.05 | 12.91 | 12.66 | 12.50 |
| SqueezeNet-v1.0 | 0.94 | 0.97 | 1.05 | 13.25 | 12.89 | 12.70 |
| SqueezeNet-v1.1 | 0.94 | 0.99 | 1.07 | 7.38 | 7.20 | 7.04 |
| VGG-11 | 1.53 | 1.55 | 1.81 | 32.56 | 30.51 | 32.27 |
| VGG-13 | 2.02 | 2.41 | 3.99 | 70.57 | 64.88 | 62.79 |
| VGG-16 | 2.41 | 3.61 | 6.02 | 91.72 | - | - |
| VGG-19 | 2.43 | 3.64 | 6.04 | 112.39 | - | - |

Table 5: Sizes of the training datasets used in the experiments

| Dataset | *mini*ImageNet100 | *mini*ImageNet | ImageNet100 | ImageNet500 | ImageNet | CUB200 |
|---|---|---|---|---|---|---|
| Size (GB) | 0.5 | 2.6 | 11.4 | 57.0 | 146.1 | 0.7 |

We define resource-constrained environments by referring to Bianco et al. (2018). The study presented investigated memory usage and computational time for classification models. The embedded system used in the study is as follows:

- NVIDIA Jetson TX1 board with 64-bit ARM® A57 CPU @ 2GHz, 4GB LPDDR4 1600MHz, NVIDIA Maxwell GPU with 256 CUDA cores

Table 4 summarizes the memory usage and computational time of the models according to the batch size for evaluation. Models found to be suitable for resource-constrained environments in the evaluation are ResNet18 (He et al. (2016)), SqueezeNet (Iandola et al. (2016)), MobileNet (Howard et al. (2017)), ShuffleNet (Zhang et al. (2018)), etc. Furthermore, we regard FSCIL models as suitable for resource-constrained environments (Zhang et al. (2021); Zhou et al. (2022); Peng et al. (2022)) since most FSCIL models employ ResNet18 as a backbone.

Table 5 shows the size of the training set of the dataset used in the experiment. NVIDIA Jetson TX1 has 16GB of internal storage. When the embedded system has no additional storage, the appropriate datasets for the embedded system are *mini*ImageNet100, *mini*ImageNet, ImageNet100, and CUB200.

### A.1    MODEL PARAMETERS

Table 6 displays the number of parameters when FourierAugment is applied. FourierAugment increases the number of channels, but FouireAugment minimally changes the number of parameters. In particular, we deleted the first convolution layer of ResNet18, so the model can learn with fewer parameters.

Table 6: Comparison of the number of light-weight model parameters

| Number of parameters | Original | +FA |
|---|---|---|
| ResNet18 | 11,688,512 | 11,646,092 |
| EfficientNet-lite0 | 3,499,108 | 3,499,972 |
| MobileNet-V2 | 3,504,872 | 3,505,736 |
| ShuffleNet-V2 | 2,278,604 | 2,279,252 |

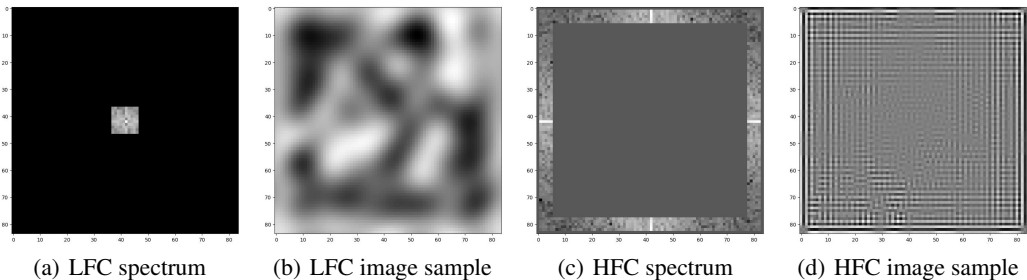

(a) LFC spectrum     (b) LFC image sample     (c) HFC spectrum     (d) HFC image sample

Figure 6: Sample LFC images and HFC images used for empirical validation.

# B  EMPIRICAL VALIDATION

We validate the characteristics of learned frequency components in resource-constrained environments with more diverse models: MobileNet-v1 (Howard et al. (2017)), ShuffleNet (Zhang et al. (2018)) and EfficientNet-lite0 (Tan & Le (2019)).

**Datasets**. In the experiment, we extract LFC images and HFC images by filtering specific frequencies from *mini*ImageNet. For LFC images, we transform the original images into the frequency domain using the Discrete Fourier Transform (DFT) and apply a filter that allows only LFC components to pass through. Fig. 6(a) represents the filtered LFC spectrum obtained from this process. Inverse transforming the LFC spectrum yields the filtered image shown in Fig. 6(b). Similarly, we obtain the HFC images by filtering the HFC spectrum, as depicted in Fig. 6(c), and then applying the inverse transformation to obtain the image shown in Fig. 6(d), which contains only HFC components. In this experiment, we create a dataset using the top $12.5\%$ of both LFC and HFC components.

**Settings**. For the three datasets used in the experiment, we limit the number of data to 500, 100, 50, 25, 10, and 5 per class. The model receives a limited number of training samples and we evaluate the models with a test dataset created in the same way. The test dataset has 100 data per class in all experiments.

**Results and Analysis**. Figs. 7, 8 and 9 delineate the result of training MobileNet, ShuffleNet, and EfficientNet-lite0 with a limited number of data per class, respectively. These experiments demonstrate that the constrained models learn less HFC as the number of data decreases—corroborating our research hypothesis over various lightweight models. Especially, the learning curves of the model learned with the original images and that with LFC converge to the same point as the number of data gets constrained under 10 or 25.

# C  EXPERIMENTS

In this section, we describe detailed experiment settings and present additional experiment results.

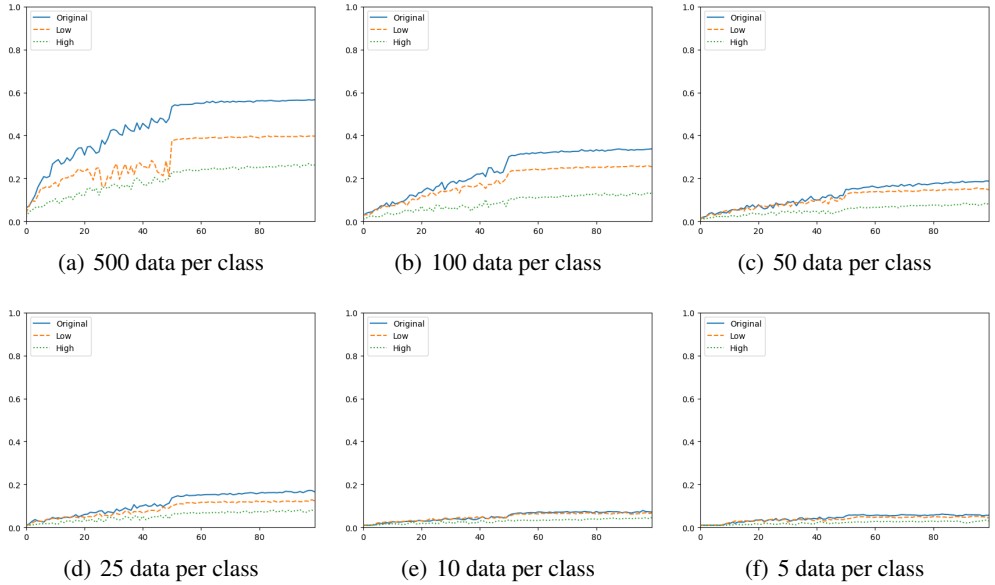

Figure 7: The effect of the number of data per class and the input frequency components on accuracy in the case of MobileNet. The blue solid line is the accuracy of the model learned from the original images. The orange dotted line and the green dotted line are the accuracies of the models learned with LFC images and HFC images, respectively.

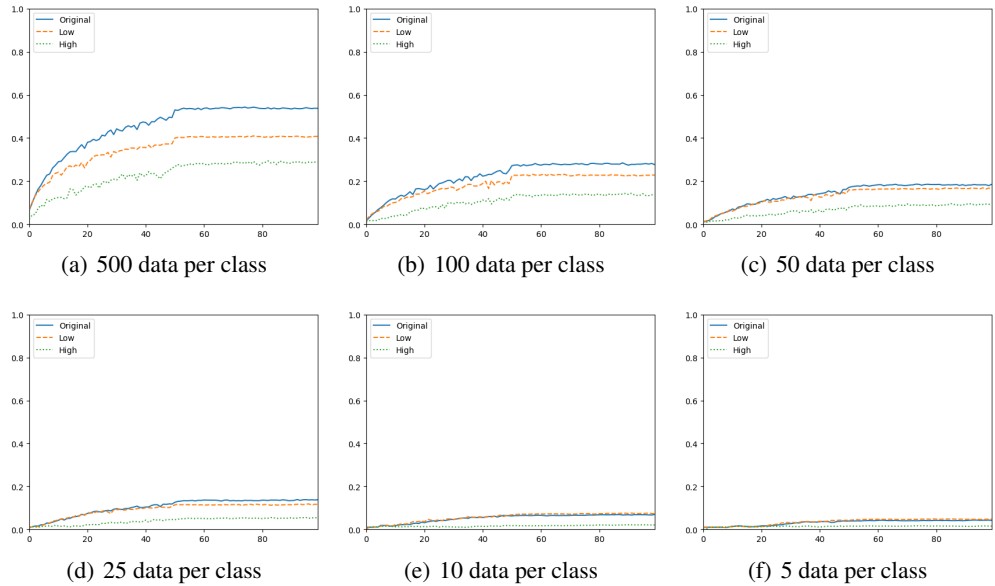

Figure 8: The effect of the number of data per class and the input frequency components on accuracy in the case of ShuffleNet. The blue solid line is the accuracy of the model learned from the original images. The orange dotted line and the green dotted line are the accuracies of the models learned with LFC images and HFC images, respectively.

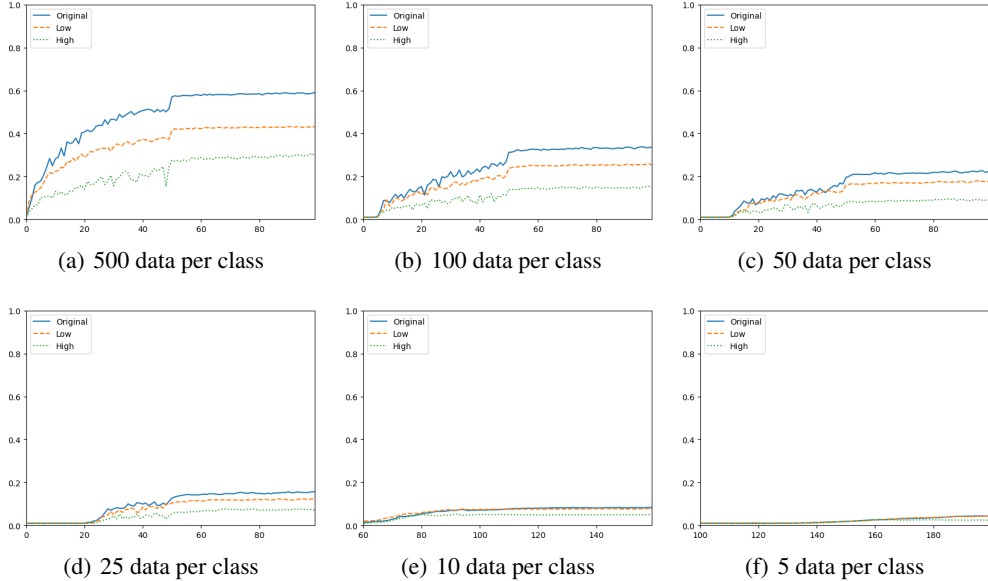

Figure 9: The effect of the number of data per class and the input frequency components on accuracy in the case of EfficientNet-lite0. The blue solid line is the accuracy of the model learned from the original images. The orange dotted line and the green dotted line are the accuracies of the models learned with LFC images and HFC images, respectively.

Table 7: Top-1 accuracies of AM, RA, DAA, and FA on EfficientNet-lite0

| Method | *mini*ImageNet-100 | *mini*ImageNet-500 | ImageNet-100 | ImageNet-500 |
|---|---|---|---|---|
| baseline | 33.80 | 59.04 | 27.90 | 55.88 |
| + AM | 33.51 | 58.99 | 36.88 | 61.15 |
| + RA | 27.74 | 56.11 | 32.46 | 59.04 |
| + DAA | 26.12 | 54.94 | 35.73 | 60.43 |
| + FA | **34.48** | **59.35** | **36.94** | **61.46** |

## C.1 IMAGE CLASSIFICATION

### C.1.1 IMPLEMENTATION DETAILS

**Computational Environment**. For all experiments, we used eight NVIDIA Tesla A30 (24G) GPUs and two Intel Xeon Gold (16/32) CPUs. We implemented FourierAugment with the Pytorch library.

**FourierAugment**. We have experimentally discovered that using two or three filters for Fourier-Augment offers optimal performance. In fact, using two or three filters does not result in a huge performance difference. Thus, we have used two channels converting the input data into a 6-channeled one. Besides, we stack the channels in increasing order from low- to high frequencies; we tested the effect of the stacking order of channels and confirmed that the stacking order does not affect the performance notably.

### C.1.2 COMPARATIVE STUDY

**Settings**. We employ EfficientNet-lite0 for additional classification experiments. We use subsets of ImageNet and *mini*ImageNet with 500 and 100 images per class. ImageNet offers with the resolution of $224 \times 224$, and *mini*ImageNet offers with the resolution of $84 \times 84$. We also measured the effectiveness of FourierAugment in unrestricted situations, using ResNet50 with *mini*ImageNet and ImageNet.

Table 8: Top-1 accuracies of RA and FA on ResNet50

| Method | $mini$ImageNet | ImageNet |
|---|---|---|
| baseline | 71.66 | 76.30 |
| +RA | 71.81 | **77.85** |
| +FA | **80.80** | 75.61 |

**Results and Analysis**. Table 7 shows the comparative study results of FourierAugment, AugMix, RandAugment, and Deep AutoAugment for each dataset. FourierAugment outperforms other data augmentation methods in the case of EfficientNet-lite0, a model suitable for resource-constrained environments. Similar to the comparative study results with ResNet18, the proposed FourierAugment displays superior performance over other data augmentation methods in all experiment conditions.

Table 8 shows the results of unconstrained environments. FourierAugment enhances the performance on mini-ImageNet (small dataset) and results in no significant performance improvement on ImageNet (large dataset). We analyze that this is because ResNet50 does not learn sufficiently various frequency components from small datasets requiring FourierAugment for enhancement, but it can learn sufficiently various frequency components from large datasets—corroborating the effectiveness of FourierAugment in resource-constrained environments.

## C.2 FSCIL

### C.2.1 PROBLEM SETTING

FSCIL includes two types of learning sessions: a single base session and a set of incremental sessions.

**Base Session**: In the base session, the FSCIL model is provided with the training dataset $\mathcal{D}^0\text{train} = (\mathbf{x}i^0, y_i^0)i = 1^{|D^0\text{train}|}$. $\mathbf{x}i^0$ represents an input image, and $yi^0$ represents the corresponding ground-truth class label. The training dataset for the base session contains a sufficient number of data samples, and the learning algorithm evaluates its performance on the test dataset $\mathcal{D}^0\text{test} = (\mathbf{x}i^0, y_i^0)i = 1^{|D^0\text{test}|}$.

**Incremental Sessions**: Following the base session, the learning algorithm proceeds with a series of incremental sessions. During these sessions, the algorithm encounters a sequence of datasets $\mathcal{D}^1, ..., \mathcal{D}^s, ..., \mathcal{D}^N$, where $\mathcal{D}^s = (\mathcal{D}^s\text{train}, \mathcal{D}^s\text{test})$, and $N$ denotes the number of incremental sessions. The label sets of object classes in each incremental session are non-overlapping, meaning that $\mathcal{C}^i \cap \mathcal{C}^j = \varnothing$ for all $i$ and $j$ where $i \neq j$. $\mathcal{C}^s$ represents the set of class labels for the $s$-th session. Furthermore, the training datasets for the incremental sessions contain an insufficient number of data samples. The $N$-way $K$-shot setting indicates that each incremental session consists of $N$ object classes, with $K$ samples per class. The evaluation protocol carried out after the final incremental session takes into account all object classes $\mathcal{C}^0 \cup \mathcal{C}^1 \cup ... \cup \mathcal{C}^N$.

### C.2.2 IMPLEMENTATION DETAILS

**Backbone**. FSCIL models used in the experiments adopted ResNet18 as the backbone. We did not make much change to the existing models to focus only on the influence of our proposed method. Specifically, we kept the training procedure the same but removed the first convolutional layer as mentioned before. For CEC and FACT, there exists a max-pooling layer following the first convolutional layer and it would reduce the raw input image resolution. We observed that this max-pooling layer harmed the performance on $mini$ImageNet and we assumed that it was due to the difficulty in utilizing frequency information from low-resolution images. The experiment on CUB200 did not exhibit a performance drop because the images have a relatively higher resolution than those of $mini$ImageNet. Therefore, we removed the max-pooling layer only when using CEC and FACT on $mini$ImageNet.

Table 9: Comparative study results of the proposed FourierAugment method and previous Fourier-based methods in image classification (*mini*ImageNet).

| Method | *mini*ImageNet-100 | *mini*ImageNet-500 |
|---|---|---|
| Baseline | 35.60 | 61.52 |
| +Xu et al. (2021) | 32.65 | 58.11 |
| +Anaya-Isaza & Zequera-Diaz (2022) | 33.00 | 52.34 |
| +Yang et al. (2022) | 32.35 | 56.37 |
| +FA | **37.35** | **65.23** |

Table 10: Comparative study results of the proposed FourierAugment method and previous Fourier-based methods in FSCIL (*mini*ImageNet).

| Method | Top-1 Accuracy in each session (%) | | | | | | | | | $PG$ |
|---|---|---|---|---|---|---|---|---|---|---|
| | 0 | 1 | 2 | 3 | 4 | 5 | 6 | 7 | 8 | |
| CEC | 78.00 | 72.89 | 69.01 | 65.45 | 62.36 | 59.09 | 56.42 | 54.28 | 52.63 | - |
| +Xu et al. (2021) | 42.58 | 38.91 | 36.14 | 34.23 | 32.06 | 30.13 | 28.38 | 27.35 | 26.10 | -26.53 |
| +Anaya-Isaza & Zequera-Diaz (2022) | 43.73 | 40.40 | 37.89 | 36.45 | 34.86 | 33.22 | 31.32 | 30.14 | 29.17 | -23.46 |
| +Yang et al. (2022) | 35.53 | 32.74 | 30.43 | 28.56 | 26.64 | 25.14 | 23.80 | 22.61 | 21.51 | -31.12 |
| + FA | 80.30 | 74.34 | 69.94 | 66.48 | 63.37 | 60.63 | 57.59 | 55.45 | **53.77** | **+1.14** |

Table 11: Ablation study results: the effect of the shape of filters on the proposed FourierAugment method (*mini*ImageNet).

| Shape of Filters | Top-1 Accuracy in each session (%) | | | | | | | | |
|---|---|---|---|---|---|---|---|---|---|
| | 0 | 1 | 2 | 3 | 4 | 5 | 6 | 7 | 8 |
| Square | 80.30 | 74.34 | 69.94 | 66.48 | 63.37 | 60.63 | 57.59 | 55.45 | **53.77** |
| + Gaussian blur | 80.77 | 74.89 | 70.23 | 66.44 | 63.05 | 59.97 | 57.04 | 54.60 | 52.84 |
| Circle | 81.22 | 75.14 | 80.79 | 67.16 | 63.56 | 60.33 | 57.50 | 55.4 | 53.65 |
| + Gaussian blur | 80.32 | 74.68 | 70.31 | 66.56 | 63.20 | 60.25 | 57.20 | 55.14 | 53.56 |

### C.2.3 COMPARATIVE STUDY

We compared FourierAugment to previous Fourier-based methods (Xu et al. (2021); Anaya-Isaza & Zequera-Diaz (2022); Yang et al. (2022)) using the image classification and the FSCIL task. In a classification task, we employ *mini*ImageNet-100 and -500 on ResNet18. Table 9 presents comparative study results of the preceding Fourier-based data augmentation techniques, revealing diminished performance. In Table 10, previous Fourier-based data augmentation methods display performance degradation; especially, other Fourier-based methods cause performance degradation by more than 20% in FSCIL.

We surmise this phenomenon occurs since previous Fourier-based methods were specifically designed towards a certain task domain, i.e., medical image processing. On the other hand, Fourier-Augment boosts performance over multiple tasks.

### C.3 ABLATION STUDY

### C.3.1 SHAPE OF FILTERS

We investigated the effect of the filter shapes on FourierAugment (Table 11): we varied the filter shapes (square or circle) and the application of the Gaussian blur. The Gaussian blur is highly inclined to degrade the performance while the filter shape does not affect the performance with a noticeable margin. We surmise that the low resolution of the images studied in the FSCIL task makes the effect of filter shape obtuse.

Table 12: Ablation study results: the effect of the number of filters on the proposed FourierAugment method (*mini*ImageNet).

| Number of Filters | Top-1 Accuracy in each session (%) | | | | | | | | |
|---|---|---|---|---|---|---|---|---|---|
| | 0 | 1 | 2 | 3 | 4 | 5 | 6 | 7 | 8 |
| CEC | 78.00 | 72.89 | 69.01 | 65.45 | 62.36 | 59.09 | 56.42 | 54.28 | 52.63 |
| + FA 6ch | 80.30 | 74.34 | 69.94 | 66.48 | 63.37 | 60.63 | 57.59 | 55.45 | **53.77** |
| + FA 9ch | 79.80 | 74.34 | 70.06 | 66.53 | 63.21 | 59.91 | 57.24 | 55.16 | 53.41 |
| + FA 12ch | 79.52 | 74.15 | 70.01 | 66.27 | 62.64 | 59.64 | 56.69 | 54.62 | 52.92 |
| + FA 15ch | 78.83 | 72.97 | 68.67 | 65.04 | 61.39 | 58.42 | 55.61 | 53.27 | 51.50 |
| + FA 18ch | 80.18 | 74.08 | 69.86 | 66.29 | 62.71 | 59.61 | 56.87 | 54.81 | 53.23 |
| + FA 21ch | 79.00 | 73.55 | 68.23 | 65.07 | 61.56 | 58.55 | 55.49 | 53.56 | 51.58 |

Table 13: Ablation study results: the effect of the first layer of ResNet18 on the proposed FourierAugment method (*mini*ImageNet)

| Method | First layer | *mini*ImageNet |
|---|---|---|
| Baseline | ∘ | 61.52 |
| +FA | ∘ | 54.48 |
| +FA | × | **65.23** |

### C.3.2 NUMBER OF FILTERS

We examined the effect of the number of filters on FourierAugment (Table 12). The result demonstrates that subdivision of the frequency bands beyond moderation degrades the performance of the model. This phenomenon is expected due to the inability to explore meaningful information with the excessive growth and granularity of the data. Excessive separation may destroy and lose essential features that rely on the whole image. Among the seven experiments, there was just one case in which the performance improved even with the increase in the number of filters. We assume that the performance improvement of the model using 6 filters and 18 channels needs further investigation.

### C.3.3 EFFECT OF THE FIRST LAYER ON RESNET18

We measured the effect of the first layer on ResNet18. Xu et al. (2020) demonstrated that the performance improves with the first convolution layer (stem layer) of ResNet getting removed when feeding frequency-related features since the transformed input functions as a few convolution layers. In a similar vein, FourierAugment includes a step of transforming images into the frequency domain, thus we delete the stem layer of ResNet. Further, we have verified the effectiveness of this elimination with ablation experiments (Table 13); the accuracy of baseline, with-the-first-layer-model, and without-the-first-layer-model corroborates the validity of our design.

In summary, the proposed FourierAugment method shows robust performance in various settings though we recommend that FourierAugment use two or three square-shaped filters for easy implementation. When applying FourierAugment to ResNet18, it is recommended to delete the stem layer.

### C.4 EFFECTIVENESS OF FOURIERAUGMENT

### C.4.1 FEATURE SPACE: T-SNE

We verify the effectiveness of FourierAugment through t-distributed stochastic neighbor embedding (t-SNE) Van der Maaten & Hinton (2008).

**Settings**. To verify qualitatively the effectiveness of FourierAugment, we compare the features learned with FourierAugment and those with other frequency-processing configurations. For this, we train ResNet18 with four data schemes (as in Sec. 5.2.1). Then, we extract the features at the last

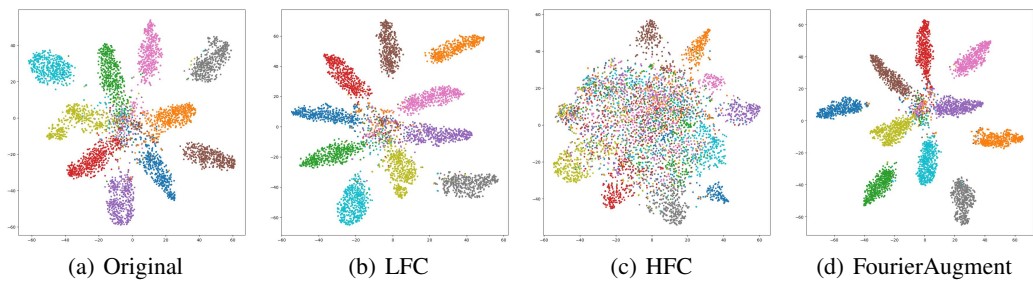

|            |            |            |            |
|:----------:|:----------:|:----------:|:----------:|
| (a) Original | (b) LFC | (c) HFC | (d) FourierAugment |

Figure 10: Variance across classes in the feature space. (a) the feature distribution by classes with the original data, (b) the feature distribution by classes with the LFC data, (c) the feature distribution by classes with the HFC data, and (d) the feature distribution by classes with FourierAugment.

block of ResNet18 (not including the fully-connected layer). The dimension of the features at the last block of ResNet18 is 512, which gets compressed to 2-dimensional vectors for visualization.

**Results and Analysis**. Fig. 10 visualizes the features learned by different models. Comparing the features learned with the original data and those with LFC reveals that intra-class variance does not display a considerable difference in the two cases (Fig. 10(a) and Fig. 10(b)). We assume that this phenomenon occurs since CNN models learn low-frequency components first (Wang et al. (2020)). Next, the features learned with HFC exhibit a huge variation across classes, *i.e., dispersed features*, (Fig. 10(c)). On the other hand, FourierAugment lets the model learn more discriminative features than other data configurations (Fig. 10(d)). Features learned with the proposed FourierAugment reveal low variance across classes. This confirms that learning each frequency component well helps improve model performance.

