# OpenReview forum: "FourierAugment: Frequency-Based Image Encoding for Resource-Constrained Vision Tasks"
_ICLR.cc/2024/Conference — Submitted to ICLR 2024_

### Official Review · Reviewer_cppX · 2023-10-28

**Soundness:** 3 good
**Presentation:** 3 good
**Contribution:** 2 fair
**Rating:** 6
**Confidence:** 3

**Summary:**

The presented method works in a specific setting of resource-constrained deep learning. The authors propose to replace the first trainable encoding block of common architectures with a manually crafted frequency-separating module. Namely, a discrete Fourier transform is applied to the input image, and the obtained frequencies are separated into several bands. Afterward, the bands are transformed into the spatial space again with the help of the inverse Fourier transform. The outputs are concatenated channel-wise and serve as an input for the neural network.

The method is evaluated both on a common classification task (with a limited number of data points per class) and on few-show class-incremental learning (FSCIL). As reported, the proposed module works better than recent augmentation strategies developed for full-scale ImageNet training such as AugMix, RandAugment, and Deep AutoAugment.

**Strengths:**

The problem considered in the paper is quite important for the community since deep learning on edge devices is rather relevant for applications. The authors clearly describe the motivation for their approach, and the description of the method is also easy to understand.
The proposed approach seems to improve the quality by a significant margin in some cases, e.g. for the ImageNet subset with 100 images per class, the accuracy is twice that of the baseline model (Tab. 1). Taking into account the simplicity of the method, this is an interesting result.

**Weaknesses:**

1. Separating the frequency bands of the network input is not novel per se: for example, it is used from time to time in generative modeling, see e.g. [1], [2]. However, usually, it is done with the help of wavelets. I recommend adding this baseline to the comparison, namely, decomposing the input image with two or three iterations of discrete wavelet transform before feeding the result to the network.
1. The authors compare their method with a number of other Fourier-based methods (Appendix C.2.3) and conclude that those methods do not perform well in their settings because they were specifically designed for medical image processing. It would be great to evaluate the proposed FourierAugment method on the medical data to demonstrate if it is competitive in that field since medical image processing often suffers from a limited amount of data.
1. Minor remarks:
    1. I find the name of the method, i.e. *FourierAugment*, misleading since it is just a way to present exactly the same information which the image contains, in a different way. In addition, it is deterministic, not random. This contradicts the way the word *augmentation* is typically used by the community nowadays.
    1. Fig. 2 contains too much empty space and does not tell a lot of useful information. For example, the horizontal axis is not very informative since there is nothing unexpected in the training dynamics, and the whole figure may be replaced with a table reporting the final performance.

[1] Hoogeboom et al. simple diffusion: End-to-end diffusion for high-resolution images. In ICML, 2023.

[2] Barron. A General and Adaptive Robust Loss Function. In CVPR, 2019.

**Questions:**

Please address the weaknesses listed above.

---

> ### Author Response · Authors · 2023-11-14
>
> 1. Wavelet
>
> As per your comment, we have added wavelet to the baseline and conducted a comparative experiment; we added 2- and 3-level decomposition. Table r3 below shows that applying wavelet decomposition to mini-ImageNet results in performance degradation.
>
> | Accuracy (%)  | mini-ImageNet 100 | mini-ImageNet 500 |
> |----------|---|---|
> | Baseline | 35.60 | 61.52 |
> | 2-level wavelet | 34.00 | 59.73 |
> | 3-level Wavelet | 34.45 | 59.59 |
> | FA | 37.35 | 65.23 |
>
> Table r3. Comparison with Wavelet.
>
> 2. Medical image
>
> We experimented on the chest X-ray dataset [1] to demonstrate that FourierAugment also works for medical images. The dataset contains 97 images and is divided into 17 disease classes. We separate 63 of 97 data into train-set and 34 into test-set, and compare FourierAugment with AugMix, RandAugment, and DeepAutoAugment using ResNet18 as in previous experiments. In Table r1, RandAugment and DeepAutoAugment were ineffective on medical datasets, AugMix achieved slightly higher performance than baseline, but FourierAugment improved performance to 29.41%.
>
> |          | Accuracy (%) |
> |----------|--------------|
> | Baseline | 20.59        |
> | AM       | 23.53        |
> | RA       | 17.65        |
> | DAA      | 20.59        |
> | FA       | **29.41**        |
>
> Table r1. Top-1 accuracies of AugMix (AM), RandAugment (RA), DeepAutoAugment (DAA), and FourierAugment (FA) on the chest X-ray dataset.
>
> 3. Name
>
> We described in the paper that our method is close to transformation, but we judged that names like Fourier-transformation could confuse the reader. The name FourierAugment explains the content and direction of the application of our research well because it can be used like any other data augmentation.
>
> 5. Fig 2
>
> As per your comment, we have tabulated the contents of Fig 2 as follows.
>
> | Accuracy (%) | 500 | 100 | 50 | 25 | 10 | 5 |
> |---|---|---|---|---|---|---|
> | Original | 61.52 | 35.60 | 25.50 | 16.17 | 10.03 | 7.12 |
> | Low | 45.68 | 26.28 | 19.98 | 13.55 | 9.85 | 7.07 |
> |High | 40.01 | 23.85 | 16.82 | 10.92 | 8.06 | 4.46 |
>
> Table r4. The horizontal axis is the number of data per class, and the vertical axis is the frequency component used.
>
> [1] TrainingData.pro. (2023). Chest X-ray - 17 Diseases [Dataset]. https://www.kaggle.com/datasets/trainingdatapro/chest-xray-17-diseases

---

> > ### Comment · Reviewer_cppX · 2023-11-20
> >
> > I thank the authors for their feedback

---

### Official Review · Reviewer_hasU · 2023-10-29

**Soundness:** 2 fair
**Presentation:** 2 fair
**Contribution:** 2 fair
**Rating:** 5
**Confidence:** 3

**Summary:**

The main contributions of the paper include the development of FourierAugment, which leverages frequency-based information to improve feature learning, and the exploration of the relationship between data quantity and frequency components learned by lightweight models. Extensive experiments demonstrate that FourierAugment outperforms baseline methods in various resource-constrained vision tasks.

**Strengths:**

1. The introduction of FourierAugment as a frequency-based image encoding method represents a novel and innovative approach in addressing resource-constrained vision tasks. FourierAugment's unique utilization of frequency components sets it apart from conventional data augmentation methods.
2. The paper's empirical study is thorough and well-designed, ensuring the validity and reliability of the results. Extensive experiments on multiple datasets and resource-constrained vision tasks provide a strong foundation for the paper's claims.
3. The paper is well-structured, with a clear presentation of the problem statement, method development, empirical study, and results.

**Weaknesses:**

The paper could benefit from a more comprehensive theoretical background and motivation section. It's important to provide a clear foundation for why FourierAugment was developed and the specific theoretical underpinnings. A deeper exploration of the relationship between frequency components and lightweight model learning could enhance the paper's overall coherence.

**Questions:**

1. The theoretical underpinnings and motivations for proposing FourierAugment, including the specific theoretical underpinnings that guided its development, would add to the coherence of the paper if they were added in full.
2. The n value is chosen empirically? Is there a definite basis for setting 2 or 3?
3. How does the method proposed in this paper compare to the comparative baseline AugMix, RandAugment and Deep AutoAugment in terms of computational efficiency and generalizability to different visual tasks?
4. Does adding the image enhancement proposed in this paper to the image classification and FSCIL tasks significantly extend the processing time of the tasks?

---

> ### Author Response · Authors · 2023-11-14
>
> 1. The theoretical underpinnings
>
> We describe in detail the motivation of our study in section 4-1. While several studies have demonstrated the significance of frequency information in vision tasks ([1], [2]), a lack of research exists pertaining to the scrutiny of lightweight models from a frequency standpoint. We hypothesized that an analytical approach focusing on frequency could ameliorate the constraints associated with lightweight models. It appears necessary to train HFC in accordance with the findings obtained from the analysis, and we propose a suitable method.
>
> 2. n value
>
> Excessive separation may destroy and lose essential features that rely on the whole image. We anticipated that using small n would improve the performance of FourierAugemnt and conducted empirical experiments on n. Table 12 displays the effect of the number of filters on FourierAugment. The result demonstrates that subdivision of the frequency bands beyond moderation degrades the performance of the model. This phenomenon is expected due to the inability to explore meaningful information with the excessive growth and granularity of the data.
>
> | Number of filters | Last session accuracy (%) |
> |----------|-----|
> | CEC | 52.63 |
> | + FA (n=2) | **53.77** |
> | + FA (n=3) | 53.41 |
> | + FA (n=4) | 52.92 |
> | + FA (n=5) | 51.50 |
> | + FA (n=6) | 53.23 |
> | + FA (n=7) | 51.58 |
>
> Table 12. Ablation study results: the effect of the number of filters on the proposed FourierAugment method (mini-ImageNet).
>
> 3. Generalizability to different visual tasks
>
> Fourier augmentation has a wide range of applications for visual tasks, like baseline methods. In particular, as demonstrated by the FSCIL experiments, FourierAugment exhibits applicability across a broader spectrum than other methods in constrained scenarios. Future research will involve the application of FourierAugment to diverse vision tasks such as detection, segmentation, and change detection.
>
> 4. Computational complexity
>
> Table 2r below shows the time complexity and memory usage of FourierAugment, AugMix, RandAugment, and DeepAutoAugment. The experiment used mini-ImageNet to measure the time and memory usage required to augment the entire dataset. Computationally, FourierAugment is faster than AugMix and DeepAutoAugment.
>
> |          | Baseline |AM|RA|DAA|FA|
> |----------|--------------|-----|-----|-----|-----|
> | Computational time (s)| 9.56  | 13.36 | 11.66 | 12.54 | 11.85 |
>
> Table 2r. Computational time of data augmentation methods.
>
> [1] Xu, Zhiqin John. "Understanding training and generalization in deep learning by fourier analysis." arXiv preprint arXiv:1808.04295 (2018).
>
> [2] Wang, Haohan, et al. "High-frequency component helps explain the generalization of convolutional neural networks." Proceedings of the IEEE/CVF conference on computer vision and pattern recognition. 2020.

---

> > ### Comment · Reviewer_hasU · 2023-11-22
> >
> > I have read all the feedback from the author, and my rating remains unchanged.

---

### Official Review · Reviewer_myuo · 2023-10-31

**Soundness:** 3 good
**Presentation:** 3 good
**Contribution:** 2 fair
**Rating:** 5
**Confidence:** 3

**Summary:**

This paper introduces FourierAugment, a frequency-based image encoding method, to address the challenges faced in resource-constrained vision tasks. By effectively utilizing frequency components from Fourier analysis, FourierAugment enables lightweight models to learn richer features with limited data.

**Strengths:**

1.  It is meaningful that the authors focus on the scenario where both the training data and computational resources are constrained.

2. The effectiveness of the proposed method on multiple resource-constrained vision tasks is demonstrated.

**Weaknesses:**

1. Learning both low- and high-frequency information in balance is not novel. There have been many works that promote uniformly learning various frequency domains by randomly masking frequency segments, but the authors did not analyze or compare their work to these related works.

[1] Stochastic Frequency Masking to Improve Super-Resolution and Denoising Networks. ECCV 2020.

[2] FSDR: Frequency Space Domain Randomization for Domain Generalization. CVPR 2021

[3] Spectrum Random Masking for Generalization in Image-based Reinforcement Learning. NeurIPS 2022.

[4] MASKED FREQUENCY MODELING FOR SELF-SUPERVISED VISUAL PRE-TRAINING. ICLR 2023.

2. I am curious if other frequency domain augmentation methods can also enhance performance, such as perturbing amplitudes[5], or randomly masking frequencies in the frequency domain[3].

[5] Proportional Amplitude Spectrum Training Augmentation for Synthetic-to-Real Domain Generalization. ICCV 2023.

3. Can the authors provide evidence that the performance improvement is not solely due to the increased parameters, considering that the number of input channels has been changed to nx3? Could you discuss the computational complexity of your proposed method compared to other methods?

4. Can the authors switch the backbone to demonstrate that there is indeed an increase in learning high-frequency features in the early stages, rather than it being a proprietary feature of ResNet?

5. It would be helpful to understand if there are any limitations to your method. Are there specific scenarios or datasets where your approach may not perform as well as existing methods? If so, what are the reasons for this?

**Questions:**

Please refer to the weaknesses.

---

> ### Author Response · Authors · 2023-11-14
>
> 1. Masking frequency segments
>
> Unlike the masking-based frequency segments ([1], [2], [3], [4]), our method uses the entire frequency and is designed for improving the performance of lightweight models‒targeting a different research goal. Furthermore, we conducted a comparative study with other frequency-based data augmentation techniques (Table 9) and confirmed that techniques developed for other tasks performed poorly in general classification.
>
> 2. Increased parameters
>
> Table 6 displays the number of parameters when different models adopt FourierAugment. FourierAugment increases the number of channels, but FouireAugment minimally changes the number of parameters. In particular, we deleted the first convolution layer of ResNet18, thus the model can learn with fewer parameters. Therefore, the performance improvement is not solely due to an increase in parameters.
>
> | Number of parameters | Original | +FA |
> |---|---|---|
> | ResNet18 | 11,688,512 | 11,646,092 |
> | EfficientNet-lite0 | 3,499,108 | 3,499,972 |
> | MobileNet-V2 |  3,504,872 | 3,505,736 |
> | ShuffleNet-V2 | 2,278,604 | 2,279,252 |
>
> Table 6. Comparison of the number of lightweight model parameters
>
> 3. Computational complexity
>
> Table r2 below shows the time complexity and memory usage of FourierAugment, AugMix, RandAugment, and DeepAutoAugment. The experiment used mini-ImageNet to measure the time and memory usage required to augment the entire dataset. Computationally, FourierAugment is faster than AugMix and DeepAutoAugment.
>
> |          | Baseline | AM | RA | DAA | FA |
> |----------|---|---|---|---|---|
> | Computational time (s) | 9.56  | 13.36 | 11.66 | 12.54 | 11.85 |
>
> Table r2. Computational time of data augmentation methods.
>
> 4. Switch the backbone
>
> We employ EfficientNet-lite0 for additional experiments. Table 7 shows the comparative study results of EfficientNet-lite0. FourierAugment outperforms other data augmentation methods in the case of EfficientNet-lite0, a model suitable for resource-constrained environments. Similar to the comparative study results with ResNet18, the proposed FourierAugment displays superior performance over other data augmentation baseline methods in all experiment conditions.
>
> 5. Limitations to FourierAugment
>
> Table 8 shows the results of unconstrained environments. FourierAugment enhances the performance on mini-ImageNet (small dataset) and results in no significant performance improvement on ImageNet (large dataset). We analyze that this is because ResNet50 does not learn sufficiently various frequency components from small datasets requiring FourierAugment for enhancement, but it can learn sufficiently various frequency components from large datasets—corroborating the effectiveness of FourierAugment in resource-constrained environments.
>
> | Method | mini-ImageNet | ImageNet |
> |---|---|---|
> | Baseline | 71.66 | 76.30 |
> | +RA | 71.81 | 77.85 |
> |+FA | 80.80 | 75.61 |
>
> Table 8. Top-1 accuracies of RA and FA on ResNet50
>
> [1] Stochastic Frequency Masking to Improve Super-Resolution and Denoising Networks. ECCV 2020.
>
> [2] FSDR: Frequency Space Domain Randomization for Domain Generalization. CVPR 2021.
>
> [3] Spectrum Random Masking for Generalization in Image-based Reinforcement Learning. NeurIPS 2022.
>
> [4] MASKED FREQUENCY MODELING FOR SELF-SUPERVISED VISUAL PRE-TRAINING. ICLR 2023.
>
> [5] Proportional Amplitude Spectrum Training Augmentation for Synthetic-to-Real Domain Generalization. ICCV 2023.

---

> > ### Comment · Reviewer_myuo · 2023-11-22
> >
> > I appreciate the author's detailed response. Nonetheless, I have a concern: The masking-based frequency segments ([1], [2], [3], [4]) are designed to ensure that the network engages the entire frequency range. However, it seems that the methods compared in Table 9 do not incorporate this kind of method. Consequently, I suggest that a more fitting comparison be made, one that takes this particular method into consideration.

---

### Official Review · Reviewer_2Mv7 · 2023-11-04

**Soundness:** 3 good
**Presentation:** 3 good
**Contribution:** 3 good
**Rating:** 5
**Confidence:** 4

**Summary:**

The paper illuminates the relationship between the learning frequency of lightweight models and the quantity of training data, presenting a clear correlation. It further introduces a novel data augmentation technique that leverages filters within the image frequency domain. The effectiveness of this new augmentation method is then demonstrated through its application in various image classification tasks, showcasing its potential to enhance model performance.

**Strengths:**

The primary strength of this paper lies in its introduction of a novel data augmentation technique. This new method stands out due to its simplicity and effectiveness, providing a valuable contribution to the field of image classification

**Weaknesses:**

The paper's weakness lies in its limited scope: the novel data augmentation method is only tested on lightweight models and small datasets. Its efficacy in broader vision tasks such as detection and segmentation remains unexplored, suggesting an area ripe for further research.

**Questions:**

Could you please clarify the meaning of "I" in formulation (1)?
Additionally, the operational details of the filter remain unclear to me.

---

> ### Author Response · Authors · 2023-11-14
>
> 1. Apply to other vision tasks
>
> We would like to thank your opinion on other vision tasks. We are working on applying FourierAugment to various vision tasks such as detection, segmentation, and change detection in future research. Furthermore, we experimented with the chest X-ray dataset [1] to demonstrate that FourierAugment also works for medical images. The dataset contains 97 images and is divided into 17 disease classes. We separate 63 of 97 data into train-set and 34 into test-set, and compare FourierAugment with AugMix, RandAugment, and DeepAutoAugment using ResNet18 as in previous experiments. In Table r1, RandAugment and DeepAutoAugment were ineffective on medical datasets, AugMix achieved slightly higher performance than baseline, but FourierAugment improved performance to 29.41%.
>
> |          | Accuracy (%) |
> |----------|--------------|
> | Baseline | 20.59        |
> | AM       | 23.53        |
> | RA       | 17.65        |
> | DAA      | 20.59        |
> | FA       | **29.41**        |
>
> Table r1. Top-1 accuracies of AugMix (AM), RandAugment (RA), DeepAutoAugment (DAA), and FourierAugment (FA) on the chest X-ray dataset.
>
> 2. Formulation (1)
>
> 'I' stands for an image data. The filter only passes the frequency component corresponding to f_i, and leaves only that frequency component in the image. This serves to divide the frequency components of the image.
>
> [1] TrainingData.pro. (2023). Chest X-ray - 17 Diseases [Dataset]. https://www.kaggle.com/datasets/trainingdatapro/chest-xray-17-diseases

---

### Meta-Review · Area_Chair_d11R · 2023-12-05

**Metareview:**

The paper presents an approach to data augmentation using the frequency domain. The authors use Fourier Domain for data augmentation and test their method on data scarce benchmarks using lightweight models.
Majority of the reviewers agree that the paper’s novelty is limited and that the scope of inquiry is also limited. Even the most positive reviewer wasn’t ready to champion the paper’s acceptance. The authors raised concerns about the reviewers not engaging in discussions which the AC noted and went over all the reviews and discussions.
Overall, the AC agrees that the novelty of the work is quite limited since there are many methods looking particularly at feeding in images using their frequency based decomposition. While this paper does find a niche in applying it to data constrained light weight model scenarios, this particular niche should be explored in a lot more detail to make the paper’s practical contributions more significant. For example, medical imaging based experiments are a claim the authors used to reply to other reviewers and only included these experiments after a reviewer asked for it. It would make the paper significantly stronger if the authors were to revise the manuscript and focus solely on these tasks where the method shines.

**Justification For Why Not Higher Score:**

I have detailed the reasons in my meta review.
Overall, the AC agrees that the novelty of the work is quite limited since there are many methods looking particularly at feeding in images using their frequency based decomposition.

**Justification For Why Not Lower Score:**

N/A

---

### Decision · Program_Chairs · 2024-01-16

Reject